# Quantitative Evaluation of Endogenous Reference Genes for RT-qPCR and ddPCR Gene Expression Under Polyextreme Conditions Using Anaerobic Halophilic Alkalithermophile *Natranaerobius thermophilus*

**DOI:** 10.3390/microorganisms13081721

**Published:** 2025-07-23

**Authors:** Xinyi Tao, Qinghua Xing, Yingjie Zhang, Belsti Atnkut, Haozhuo Wei, Silva Ramirez, Xinwei Mao, Baisuo Zhao

**Affiliations:** 1Graduate School, Chinese Academy of Agricultural Sciences, Beijing 100081, China; taoxinyi121dx@163.com (X.T.);; 2Department of Biology, College of Natural and Computational Science, Injibara University, Injibara P.O. Box 40, Ethiopia; 3Department of Civil Engineering, Stony Brook University, Stony Brook, NY 11794, USA

**Keywords:** endogenous reference genes, RT-qPCR, polyextreme condition, extremophiles, *Natranaerobius thermophilus*

## Abstract

Accurate gene expression quantification using reverse transcription quantitative PCR (RT-qPCR) requires stable reference genes (RGs) for reliable normalization. However, few studies have systematically identified RGs suitable for simultaneous high salt, alkaline, and high-temperature conditions. This study addresses this gap by evaluating the stability of eight candidate RGs in the anaerobic halophilic alkalithermophile *Natranaerobius thermophilus* JW/NM-WN-LF^T^ under combined salt, alkali, and thermal stresses. The stability of these candidate RGs was assessed using five statistical algorithms: Delta CT, geNorm, NormFinder, BestKeeper, and RefFinder. Results indicated that *recA* exhibited the highest expression stability across all tested conditions and proved adequate as a single RG for normalization in both RT-qPCR and droplet digital PCR (ddPCR) assays. Furthermore, *recA* alone or combined with other RGs (*sigA*, *rsmH*) effectively normalized the expression of seven stress-response genes (*proX*, *opuAC*, *mnhE*, *nhaC*, *trkH*, *ducA*, and *pimT*). This work represents the first systematic validation of RGs under polyextreme stress conditions, providing essential guidelines for future gene expression studies in extreme environments and aiding research on microbial adaptation mechanisms in halophilic, alkaliphilic, and thermophilic microorganisms.

## 1. Introduction

Reverse transcription quantitative PCR (RT-qPCR) is a commonly employed technique for analyzing gene expression levels. It provides high sensitivity, accuracy, and rapid results [1,2]. Nevertheless, achieving accurate and reliable RT-qPCR results critically relies on the careful selection and thorough validation of stable endogenous RGs, a process that is fundamental to the design of any new qPCR experiment [3]. RGs must remain stable across experimental conditions. Fluctuations in their expression can distort gene expression quantification. This results in flawed normalization and incorrect conclusions [4,5]. Therefore, identifying stable RGs is essential for accurate RT-qPCR data interpretation [6,7].

Although the use of RGs is well-established, previous studies have often relied on genes such as *16S rRNA* as internal controls. However, the expression stability of these commonly used RGs has not always been validated. For instance, *16S rRNA* has been shown to exhibit expression variability under certain experimental conditions [8], and its relatively high expression level compared to other housekeeping genes may interfere with accurate normalization [9]. Similar limitations have been observed with *gapdh*, which has limited value for gene expression normalization [10]. These cases underscore the importance of validating RGs under specific experimental conditions to ensure reliable and accurate normalization.

Previous studies have identified suitable RGs under specific single-stress conditions. In *Pseudomonas* sp. AU10, *recA* and *ftsZ* were identified as stable reference genes during exponential growth at both 4 °C and 30 °C and after exposure to cold shock [11]. Under pH stress conditions, with exposure to pH levels of 9.0 and 5.0, the genes *rpoB*, *rpoD*, and *fabD* were selected in *Acinetobacter baumannii* [12]. These findings underscore the importance of context-specific RG selection but are limited to isolated environmental stressors. In a prior investigation, we discovered *pdp* as a stable RG from *polC, dnaK, pyrD, recA, pdp,* and *rplY* across different salt concentrations (4%, 8%, 12%, and 16% NaCl) in *Alkalicoccus halolimnae*, a moderately halophilic bacterium [13]. However, RG stability under combined extreme conditions—high salinity, high alkalinity, and high temperature—has not been systematically assessed, highlighting a critical gap.

To address this, we focus on *Natranaerobius thermophilus*, an obligately anaerobic, halophilic alkalithermophile that grows at 2.5–4.9 M Na^+^, pH 8.3–10.1, and 35–56 °C, with optimal growth at 3.9 M Na^+^, pH 9.5, and 53 °C. Its exceptional adaptation to polyextreme environments makes it an ideal model for studying RG stability under combined stress conditions. In this study, eight housekeeping genes from *Natranaerobius thermophilus—rsmH*, *pdp*, *recA*, *accD*, *sigA*, *gyrA*, *rpoB*, and *dnaK*—as candidate RGs were selected, and their expression stability was evaluated via RT-qPCR assays using multiple analytical tools, including geNorm, NormFinder, BestKeeper, RefFinder, and the comparative Ct method. The most stable RGs were then used to normalize the expression of *proX*, *opuAC*, *mnhE*, *nhaC*, *trkH*, *ducA*, and *pimT* through RT-qPCR analysis. To independently verify the normalization reliability, the selected RGs were further validated using droplet digital PCR (ddPCR), providing an additional layer of accuracy. The findings of our study reveal that *recA* exhibited the highest stability across all tested conditions, making it an ideal candidate for gene normalization in *N. thermophilus*. Additionally, the combination of *recA* with other RGs, such as *sigA* and *rsmH*, improved the accuracy of gene expression normalization.

In summary, previous studies on endogenous RGs in extremophiles have primarily focused on single environmental stress factors such as salinity, alkalinity, or temperature. However, extremophiles naturally experience multiple coexisting stressors in their environments, leading to more complex transcriptional regulation. Systematic evaluation of reference gene stability under such combined polyextreme conditions remains scarce. To address this critical gap, our study comprehensively assessed the expression stability of eight candidate reference genes in the facultative anaerobic halophilic alkalithermophile *N. thermophilus* under combined salt, alkaline, and thermal stresses. By integrating multiple statistical algorithms along with validation methods such as RT-qPCR and ddPCR, we identified reliable reference genes suitable for gene expression normalization in complex environmental conditions.

This research identifies reliable endogenous RGs for use in *N. thermophilus*, providing a robust methodological foundation for gene expression normalization under combined extreme environmental stresses. By systematically validating RGs under simultaneous salt, alkaline, and thermal conditions using integrated RT-qPCR and ddPCR approaches, this study establishes essential technical groundwork supporting accurate gene expression analyses in extremophilic microorganisms. These findings contribute valuable methodological insights that facilitate future investigations into microbial adaptation and resilience in multifactorial extreme environments.

## 2. Materials and Methods

### 2.1. Bacterial Strains and Growth Conditions

*N. thermophilus* was isolated from a mixed water–sediment sample collected from the sediment of Lake Fazda, Wadi An Natrun, Egypt, during May 2005. At the time of collection, the lake water had a salinity of 4.7 M and a pH of 9.8 at 25 °C. It grows at 2.5–4.9 M Na^+^, a pH^55°C^ of 8.3–10.1, and temperatures ranging from 35 to 56 °C, with optimal growth occurring at 3.9 M Na^+^, pH^55°C^ 9.5, and 53 °C, respectively. The pH range for growth was determined in the medium at 55 °C [14]. The growth medium was modified from a previous study by adding varying amounts of NaCl to achieve final salinity concentrations of 3.0 M and 4.0 M Na^+^ [14]. To adjust the pH, 5.0 M HCl (CarlRoth, Karlsruhe, Germany) was used to titrate the medium to pH values of 8.6 and 9.6 at 55 °C. Cultures were incubated at 42 °C and 52 °C to collect samples under different thermal conditions. Cultures of *N. thermophilus* were incubated at 42 °C and 52 °C to represent distinct points within the organism’s viable growth temperature range (35–56 °C). The optimal growth temperature is approximately 53 °C; 52 °C was selected to closely approximate this while ensuring stable and reproducible cultivation conditions.

To enhance ecological relevance, the selected salt concentrations and pH values were designed to approximate the natural habitat conditions of strain JW/NM-WN-LF^T^, as reflected by the lake’s salinity of 4.7 M and pH 9.8 at the time of isolation. This experimental design aims to simulate the natural environmental conditions encountered by the microorganism in situ, thereby increasing the applicability and ecological validity of our findings. Bacterial inocula (1:20 dilution) were transferred into anaerobic media under the six conditions described above. Growth was monitored by measuring optical density at 600 nm (OD_600_) with a HACH DR2800 spectrophotometer. Samples were collected when the cultures reached an OD_600_ of 0.5. Samples of 1.5 mL were centrifuged at 13,000 rpm for 5 min at 4 °C. Cell pellets were preserved in RNA stabilization reagent (Qiagen, Hilden, Germany) and stored at −80 °C for subsequent RNA extraction. Each experiment was performed with three independent biological replicates and three technical replicates.

### 2.2. Total RNA Extraction and cDNA Synthesis

RNA extraction was performed using a Bacterial RNA Kit (Omega, Norwalk, CT, USA), and concentration was assessed with a NanoDrop ND-1000 spectrophotometer (NanoDrop Technologies Inc., Wilmington, DE, USA). RNA concentration ranged from 300 to 400 ng/μL, with A260/A280 ratios between 1.8 and 2.0 and A260/A230 ratios around 2.0, indicating high purity and minimal contamination. The cDNA synthesis was performed using 100 ng of RNA. Reverse transcription was performed according to the manufacturer’s instructions using a PrimeScript RT Reagent Kit with gDNA Eraser (Takara, Tokyo, Japan). The synthesized cDNA was dissolved in RNase-free water and stored at −20 °C for later analysis. The experiment was conducted with three independent biological replicates, each with three technical replicates.

### 2.3. Selection of Candidate Genes and Primer Design

Eight candidate RGs were selected from commonly used housekeeping genes based on previous literature and the genomic information of *N. thermophilus*. Primers for these genes were designed using Primer Premier 5.0, referencing the *N. thermophilus* genome sequence (NZ_CP144221.1; https://www.ncbi.nlm.nih.gov/nuccore/NZ_CP144221.1, accessed on 9 February 2025) and synthesized by Sangon Biotech in Shanghai, China. The selected candidate RGs included *rsmH* (Nther_1296, ACB84879.1), encoding a ribosomal RNA small subunit methyltransferase involved in rRNA methylation to ensure accuracy and efficiency of protein synthesis; *pdp* (Nther_1665, ACB85239.1), encoding pyrimidine nucleoside phosphorylase, which participates in nucleotide metabolism maintaining nucleic acid homeostasis; *recA* (Nther_1473, ACB85056.1), encoding recombinase A, a key protein in DNA repair and homologous recombination essential for genomic stability; *accD* (Nther_0846, ACB84431.1), encoding acetyl-CoA carboxylase subunit β/α, involved in fatty acid biosynthesis; *sigA* (Nther_1215, ACB84798.1), encoding RNA polymerase sigma factor A, which regulates transcription initiation and maintains basal transcription levels; *gyrA* (Nther_0008, ACB83607.1), encoding DNA gyrase subunit A, involved in DNA supercoiling regulation; *rpoB* (Nther_0186, ACB83785.1), encoding a DNA-dependent RNA polymerase β subunit, catalyzing RNA synthesis; and *dnaK* (Nther_1183, ACB84766.1), encoding a molecular chaperone protein involved in protein folding and stress responses. In addition, seven functional target genes—*proX* (Nther_1620, ACB85194.1), part of an ABC transporter system involved in compatible solute uptake for osmotic stress adaptation; *opuAC* (Nther_0728, ACB84318.1), a component of osmoprotectant uptake systems facilitating transport of compatible solutes such as glycine betaine; *mnhE* (Nther_0501, ACB84097.1), a subunit of a multisubunit Na^+^/H^+^ antiporter involved in sodium ion homeostasis and pH regulation; *nhaC* (Nther_0736, ACB84326.1), a Na^+^/H^+^ antiporter protein involved in sodium efflux and pH balance; *trkH* (Nther_0103, ACB83702.1), a potassium uptake protein critical for intracellular ion homeostasis under salt stress; *ducA* (Nther_2657, ACB86212.1), a predicted amino acid transporter potentially involved in nutrient uptake and metabolism; and *pimT* (Nther_0279, ACB83877.1), a putative transporter associated with metabolite transport and stress response—were selected for further gene expression analysis. These target genes corresponded to proteins that were consistently upregulated under salt, alkaline, and thermal stress conditions, as identified through iTRAQ-based quantitative proteomic analysis of *N. thermophilus* [13,15]. The primer sequences used for RT-qPCR and ddPCR assays are provided in Appendix A.

### 2.4. RT-qPCR Efficiency and Assays

A temperature gradient PCR (ranging from 54 °C to 60 °C) was performed for each primer to identify the optimal annealing temperature. Diluted cDNA was then used as a template for RT-qPCR, performed in triplicate with the following dilution factors: 1, 10, 100, 1000, 10,000, and 100,000-fold. The specificity and amplification efficiency of the primers were evaluated by generating a relative standard curve. The RT-qPCR efficiency (E) for each primer pair was determined based on the regression coefficients (*R*) obtained from the linear regression analysis [3], using the following equation:E(%) = (10^−(1/*R*)^ − 1) × 100

RT-qPCR was performed using the CFX96™ Real-Time PCR Detection System (Bio-Rad, Hercules, CA, USA) and Power SYBR^®^ Green PCR Master Mix (Applied BioSystems, Waltham, MA, USA). The cDNA RT-PCR products were diluted to 20 ng/μL and used as templates. Each reaction was prepared in a final volume of 20 μL, consisting of 10 μL SYBR^®^ Premix Ex Taq^®^ II, 0.4 μL of each forward and reverse primer (10 μmol/L), 2 μL of cDNA (20 ng/μL), and 7.2 μL of DNase/RNase-free water (Invitrogen™, Waltham, MA, USA). The reaction mixture was assembled on ice. The RT-qPCR program consisted of an initial denaturation at 95 °C for 30 s, followed by 10 s of denaturation at 9 °C5 °C, primer-specific annealing for 30 s, extension at 72 °C for 30 s, and a total of 40 amplification cycles. Melting curve analysis was carried out between 65 °C and 95 °C, with a 0.5 °C increase per second. Each primer pair was tested in triplicate, accompanied by three no-template controls (NTC).

### 2.5. Stability of Gene Expression and Minimum Number of RGs

Eight candidate genes were assessed under different experimental conditions to verify the reliability and accuracy of the selected RGs for data normalization. To investigate gene expression characteristics under combined salt, alkali, and thermal stresses, samples were collected from cultures grown under each individual condition and analyzed via RT-qPCR. Subsequently, expression data from these discrete conditions were integrated to simulate transcriptional responses under multifactorial extreme stress. Gene expression stability was evaluated using statistical software tools such as geNorm (https://seqyuan.shinyapps.io/seqyuan_prosper/, accessed on 25 February 2025), NormFinder (https://seqyuan.shinyapps.io/seqyuan_prosper/, accessed on 25 January 2025), BestKeeper (http://blooge.cn/RefFinder/?type=reference, accessed on 25 January 2025), and RefFinder (http://blooge.cn/RefFinder/?type=reference, accessed on 23 January 2025). For RT-qPCR, raw Ct values were processed using BestKeeper, while data from geNorm and NormFinder were converted into linear values based on the lowest Ct of each gene. Gene stability was evaluated via BestKeeper’s standard deviation (SD), along with the stability values (M) from geNorm and stability values (SVs) from NormFinder. The optimal number of RGs for normalization was determined using geNorm’s pairwise variation (V). RefFinder, a web-based analytical tool, integrates four algorithms: Delta CT, BestKeeper, geNorm, and NormFinder, to rank candidate RGs. RefFinder assigns a weight to each gene according to its ranking in each program and then calculates the geometric mean of these weights to determine the overall ranking.

### 2.6. Validation of the Selected RGs

The expression of the *proX*, *opuAC*, *mnhE*, *nhaC*, *trkH*, *ducA*, and *pimT* genes in *N. thermophilus* was used to validate the selected RGs. The expression levels of these seven target genes under various extreme conditions were normalized relative to *recA* by comparing their Ct values. Four optimal RG combinations, based on the geometric mean of the Ct values, were identified: *recA* and *sigA*, *recA* and *rsmH*, *recA* and *accD*, and *recA* and *pdp*. Relative gene expression levels were determined using the comparative 2^−ΔCt^ method. RT-qPCR was used to determine the expression patterns of the seven target genes under different stress conditions (salinity, alkalinity, and temperature), which were then normalized to combinations of *recA* and the selected RGs. Protein expression levels served as a benchmark to assess the effectiveness of the chosen RGs for data normalization. Furthermore, the expression profile of each target gene under different conditions was assessed using ddPCR and normalized relative to *recA*. The relative copy number of each target gene was determined by dividing the absolute copy number of the target gene by the absolute copy number of the RG, as analyzed using ddPCR.

## 3. Results

### 3.1. Amplification Efficiency of Candidate RGs

Eight candidate RGs were selected for analysis: ribosomal RNA small subunit methyltransferase H (*rsmH*), pyrimidine-nucleoside phosphorylase (*pdp*), recombinase A (*recA*), acetyl-coenzyme A carboxylase carboxyl transferase subunits beta/alpha (*accD*), RNA polymerase sigma factor SigA (*sigA*), DNA gyrase subunit A (*gyrA*), DNA-directed RNA polymerase subunit beta (*rpoB*), and chaperone protein DnaK (*dnaK*), as detailed in the Appendix A. An optimal annealing temperature of 58 °C was determined for all primers. Standard PCR amplification confirmed the specificity of the primers, with each RG generating a single, clear band of the expected size, free of primer-dimer formation (Figure 1). Sanger sequencing further verified that the amplified fragments matched the sequences in the NCBI database. Melting curve analysis showed a distinct single peak for each candidate RG (Figure 2). Table 1 summarizes the RT-qPCR performance parameters for each RG, including the slope, correlation coefficient (R^2^), and amplification efficiency. Amplification efficiencies ranged from 90.3% (*rpoB*) to 108.7% (*rsmH*), with R^2^ values between 0.986 and 0.999, indicating strong linear correlations. These results confirm that the primers exhibited high specificity and sensitivity, making them suitable for subsequent quantitative analyses.

### 3.2. Candidate RGs Expression Levels

An ideal RG should maintain stable expression levels similar to those of the target genes under varying experimental conditions. The distribution of raw Ct values for the candidate RGs is shown in Figure 3, ranging from 20 to 35 across all conditions. *RecA*, *pdp*, and *gyrA* exhibited relatively narrow Ct value ranges, suggesting more stable expression. According to Delta Ct analysis, *recA* demonstrated the lowest standard deviation, indicating the highest stability across different salt, alkaline, and temperature conditions. In contrast, *dnaK* exhibited the greatest fluctuation in Ct values, suggesting it was the least stable RG under varying stress conditions.

### 3.3. Expression Stability of Candidate RGs

Ct values were analyzed using specialized algorithms such as geNorm, NormFinder, BestKeeper, and RefFinder (Table 2). These analyses identified the most stable RGs for each condition and determined the optimal number of RGs necessary for normalization (Figure 4).

### 3.4. GeNorm Analysis

Genes with M-values below 1.5 are considered to have stable expression, with lower M-values reflecting higher stability. All eight candidate RGs had M-values below 0.55 (Table 2), demonstrating high stability. The most stable pairs identified via geNorm were *rsmH*/*recA* (varying salinity), *pdp*/*sigA* (heat stress), and *pdp*/*recA* (alkalinity stress). *AccD*, *dnaK*, and *rpoB* were the least stable under different conditions. Further analysis indicated that *recA* and *sigA* had the most stable expression under combined salt, alkaline, and temperature stress conditions, with an M-value of 0.18 (Figure 5a). Although *rpoB* ranked lowest in stability, its M-value remained below the 1.5 threshold.

Pairwise variation (Vn/n + 1) analysis revealed that all V2/3 values were below 0.15 (Figure 4), indicating that two internal RGs are sufficient for normalization.

### 3.5. NormFinder Analysis

NormFinder, a Microsoft Excel-based tool, was utilized to assess the expression stability of candidate RGs. Lower SVs indicate higher gene expression stability. The SVs for *recA* under varying salinity, temperature, and pH conditions were 0.05, 1.03, and 0.5, respectively (Table 2; Figure 5b). *RecA* emerged as the most stable RG across most conditions, with the exception of varying temperature stress, where *accD* displayed the highest stability. However, *accD* showed considerable instability under varying salinity. *DnaK* was consistently ranked as the least stable RG under all tested conditions. Overall, despite slight variations under specific stresses, *recA* demonstrated the highest overall stability across combined polyextreme environments in *N. thermophilus*.

### 3.6. BestKeeper Analysis

BestKeeper software ranks RG stability based on standard deviation (SD) and coefficient of variation (CV) using raw Cq values. Stability was inversely related to SD. Table 2 shows the stability analysis results of the eight candidate RGs, ranked from highest to lowest under various stress conditions. Considering all samples together, the *gyrA* gene was identified as the most stable internal RG (Figure 5c). However, when individual stress conditions such as salinity, temperature, and pH were analyzed separately, *gyrA* was not consistently the most stable RG (Table 2). Specifically, *gyrA* ranked fourth in terms of stability under pH variation. In contrast, *recA* ranked first for gene expression stability under varying temperature and pH conditions.

### 3.7. RefFinder Analysis

RefFinder (http://blooge.cn/RefFinder/?type=reference, accessed on 23 January 2025), which combines the results from geNorm, NormFinder, and BestKeeper, was used to generate a comprehensive stability ranking for the candidate RGs. The RefFinder analysis confirmed that *recA* was the most suitable RG across all individual and combined stress conditions (Table 2; Figure 5d), consistent with the findings obtained from BestKeeper, geNorm, and NormFinder analyses. In addition, *recA*, *pdp*, and *gyrA* exhibited narrow Ct value ranges across all tested conditions (Figure 3), further supporting their relative stability. Gene expression stability was thoroughly evaluated by combining results from the BestKeeper, geNorm, NormFinder, and RefFinder programs (Table 2; Figure 5d). Regardless of variations in salinity, temperature, or pH levels, *recA* was consistently identified as the most suitable internal control gene for RT-qPCR gene expression analysis in *N. thermophilus*.

### 3.8. Validation of the Selected RGs

The *proX*, *opuAC*, *mnhE*, *nhaC*, *trkH*, *ducA*, and *pimT* genes are critical for glycine betaine synthesis and osmotic regulation under salt-alkali-heat stress conditions. RT-qPCR and ddPCR assays were conducted to assess the expression patterns of the selected target genes under varying salinity, temperature, and pH conditions, in order to validate the suitability of the chosen RGs.

For RT-qPCR analysis, the expression levels of the seven target genes were normalized using *recA* and four optimal RG combinations recommended by the algorithms: *recA* and *sigA*, *recA* and *rsmH*, *recA* and *accD*, and *recA* and *pdp*. The expression trends of all target genes remained consistent across the various normalization approaches, with Ct value variations of less than five cycles (Figure 6). Regardless of whether normalized to *recA* alone or to the four RG combinations, the overall expression trends of *proX*, *opuAC*, *mnhE*, *nhaC*, *trkH*, *ducA*, and *pimT* remained consistent. Under each stress condition, normalization with these internal RGs revealed uniform trends of upregulation or downregulation across the target genes, although the magnitude of fold changes varied depending on the normalization strategy (Figure 6). These results suggest that *recA* is an ideal RG, demonstrating reliable performance across salt, alkaline, and temperature stress conditions.

For ddPCR validation, gene expression levels were similarly normalized using *recA* (Figure 7). The expression patterns for most target genes were consistent between RT-qPCR and ddPCR results, although *ducA* and *nhaC* exhibited some differences. Specifically, under salt conditions of 3.0 M and 4.0 M Na^+^, these two genes were upregulated in RT-qPCR but downregulated in ddPCR (Figure 7). Nevertheless, the expression trends of the remaining target genes remained similar regardless of the method used for normalization. These findings confirm that *recA*, either alone or in combination with other RGs such as *sigA* and *rsmH*, acts as a dependable internal control for gene expression normalization and analysis under different salinity, temperature, and pH conditions in *N. thermophilus*.

## 4. Discussion

RT-qPCR remains the gold standard for gene expression analysis due to its sensitivity, specificity, and wide dynamic range [16], particularly for organisms subjected to extreme environmental stresses such as high salinity, alkalinity, and temperature. Accurate normalization of RT-qPCR data depends critically on the selection of stable RGs, making their validation under relevant conditions essential for reliable and reproducible results. However, many RGs traditionally employed under standard laboratory conditions may display variability when tested under polyextreme environments. Addressing the lack of systematic validation under such conditions, this study investigated the expression stability of candidate RGs in *N. thermophilus*, a halophilic alkalithermophile, to support accurate gene expression analysis in extreme settings.

Among the eight candidate RGs evaluated, *recA* exhibited the highest stability across combined salt, alkali, and thermal stresses, as confirmed via five independent algorithms. This finding is consistent with its biological role in maintaining genomic integrity under extreme environmental conditions through DNA repair and homologous recombination [17]. Notably, the different analytical tools applied—geNorm, NormFinder, BestKeeper, RefFinder, and the comparative Ct method—each offer complementary evaluation perspectives, thereby strengthening the reliability of the selection process. The consistent identification of *recA* across these methods highlights its robustness. Its stable expression across a broad range of conditions underscores its suitability as a single RG for RT-qPCR and ddPCR normalization in *N. thermophilus*.

While *recA* demonstrated outstanding stability in *N. thermophilus* under polyextreme conditions, previous investigations in *A. halolimnae* revealed that *recA* was not the most stable RG across varying salinities [13]. These findings collectively underscore the necessity of empirical validation of RGs for each specific organism and environmental condition, emphasizing that commonly used housekeeping genes cannot be universally applied without rigorous evaluation. Gene expression patterns are often strain-specific and condition-dependent, necessitating tailored selection strategies even for closely related species.

In addition to using *recA* alone, combining it with other RGs such as *sigA* and *rsmH* further enhanced normalization accuracy. This observation aligns with previous studies highlighting the advantage of multiple RGs for reliable normalization under complex stress conditions [18,19]. Although *recA* alone proved sufficient in *N. thermophilus*, RG combinations may provide added robustness when analyzing diverse stress intensities or more subtle shifts in gene expression profiles. Such strategies are particularly valuable when investigating multifactorial stress responses where minor expression changes must be accurately captured.

Beyond RT-qPCR, the reliability of the selected RGs was independently validated using ddPCR, providing an additional layer of verification. DdPCR enables absolute quantification without the need for standard curves and exhibits increased tolerance to PCR inhibitors, rendering it particularly suitable for gene expression studies in environmental extremophiles.

This study represents the first systematic validation of RGs under simultaneous salt, alkali, and heat stress, reflecting conditions closer to the natural habitats of extremophiles. In contrast, previous research has predominantly focused on single-stress conditions [20,21], thereby limiting their applicability to real-world environmental challenges. We found that the *recA* gene exhibited the highest expression stability across all tested conditions, a result consistently confirmed via five statistical methods: geNorm, NormFinder, BestKeeper, RefFinder, and the comparative Ct method. This stability aligns closely with the critical biological functions of *recA* in maintaining genomic integrity, DNA repair, and homologous recombination under extreme environments. Additionally, we observed that combining *recA* with other reference genes such as *sigA* and *rsmH* further enhances normalization accuracy, particularly in detecting subtle gene expression changes induced by multifactorial stresses. Our findings thus provide a valuable reference for future gene expression studies in halophilic, alkaliphilic, and thermophilic microorganisms and highlight the importance of designing normalization strategies tailored to extreme conditions.

Nevertheless, RG performance can vary substantially across species, strains, and experimental setups. As evidenced in studies on *Vibrio parahaemolyticus* and *Bradyrhizobium* USDA 110T [20,21], RG stability is not universally conserved. Therefore, preliminary validation under specific experimental conditions remains essential. Future research should assess the long-term stability of *recA* and other RGs under chronic polyextreme stresses and evaluate their applicability across a broader range of bacterial extremophiles, such as *Halomonas*, *Alkalibacterium*, *Thermus*, and *Salinivibrio* species. In addition, exploring the development of stress-specific RGs tailored to particular stress response pathways could further enhance normalization accuracy. Integrating multi-omics approaches, including transcriptomics and proteomics, may provide a more comprehensive framework for optimizing gene expression studies in diverse extremophilic bacterial systems.

## 5. Conclusions

This study systematically identified *recA* as the most stable RG for gene expression normalization in *N. thermophilus* under combined salt, alkalinity, and heat stresses. *RecA* alone, or in combination with *sigA* and *rsmH*, provided robust normalization across both RT-qPCR and ddPCR platforms. By systematically validating RGs under polyextreme environmental conditions, this work fills a critical gap in the literature and establishes a solid foundation for future gene expression analyses in extremophilic systems. These findings also offer valuable guidance for selecting stable RGs in studies on halophilic, alkaliphilic, and thermophilic bacteria, with potential applications in microbial biotechnology and environmental adaptation research.

While these findings provide a solid foundation for the future, certain limitations should be noted. First, this study was conducted using a single strain, so caution should be exercised when extrapolating these results to other extremophiles or environmental conditions. Second, the stress conditions employed primarily represent long-term exposures and do not account for potential short-term fluctuations or complex multifactorial interactions found in natural environments. Finally, although the stability of candidate reference genes was confirmed via RT-qPCR and ddPCR, incorporation of transcriptome-wide RNA-Seq data would provide a more comprehensive validation. Future studies should expand validation across multiple strains and stress types and integrate multi-omics approaches to further optimize gene expression normalization strategies in extreme environments.

## Figures and Tables

**Figure 1 microorganisms-13-01721-f001:**
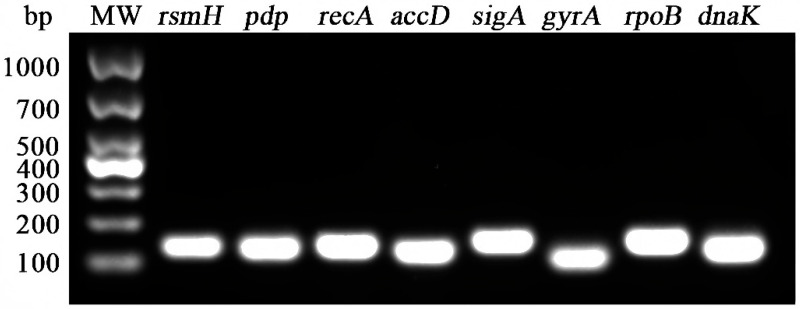
Agarose gel electrophoresis of RT-qPCR products of candidate RGs. MW: DL1000 DNA Marker, *rsmH* (133 bp), *pdp* (131 bp), *recA* (137 bp), *accD* (125 bp), *sigA* (149 bp), *gyrA* (101 bp), *rpoB* (148 bp), and *dnaK* (125 bp).

**Figure 2 microorganisms-13-01721-f002:**
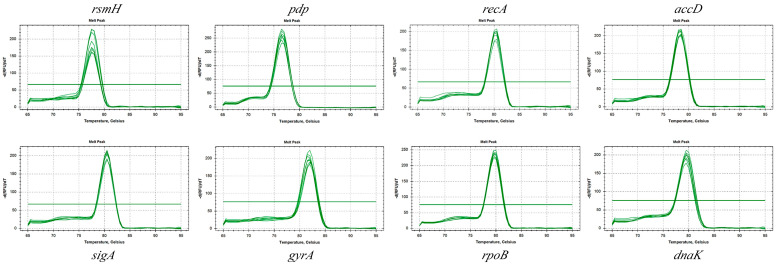
Melting curves of candidate RGs.

**Figure 3 microorganisms-13-01721-f003:**
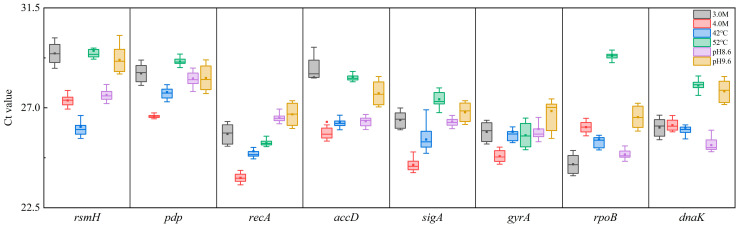
Distribution of Ct values of 8 candidate RGs within the different samples of *N. thermophilus* under different experimental conditions. Box plots of Ct distribution for each candidate RG. Box-plot elements show: box limits represent the 25 and 75 percentiles, center long line represents the median, center short line represents the mean value, and whiskers represent the 1.5I QR (interquartile range).

**Figure 4 microorganisms-13-01721-f004:**
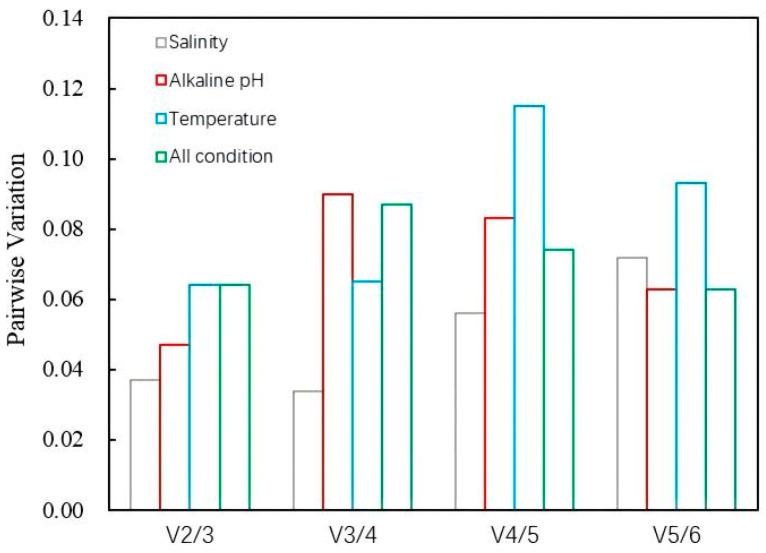
Determination of the optimal number of candidate RGs for normalization (geNorm) of *N. thermophilus* under different salinities, pH values, temperatures, and combined all conditions. The value of the abscissa indicates the number of RGs.

**Figure 5 microorganisms-13-01721-f005:**
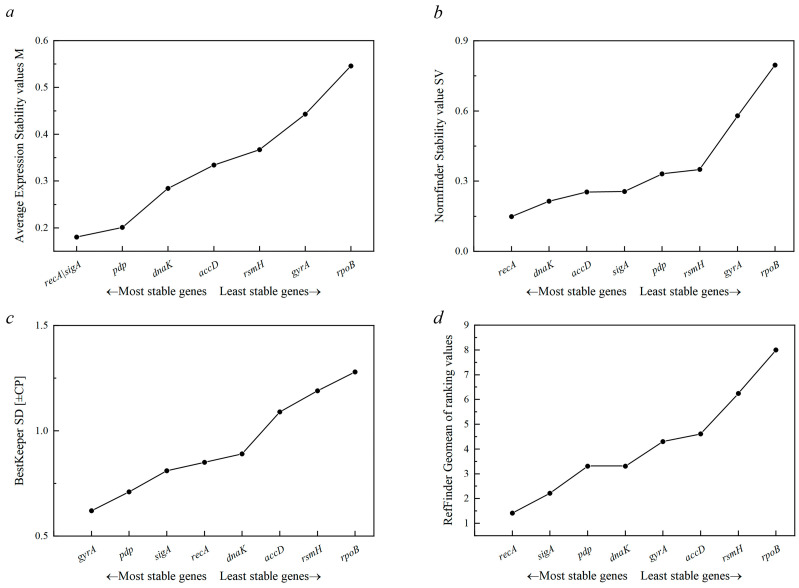
Expression stability of candidate RGs in *N. thermophilus* in response to saline-alkali heat stress according to cycle threshold (Ct) values from RT-qPCR. RGs were ranked via geNorm (**a**), NormFinder (**b**), BestKeeper (**c**), and RefFinder (**d**) programs. The lower M-value (geNorm), stability value SV (NormFinder), standard deviation (BestKeeper), or geomean values (RefFinder) for a certain gene indicates more stable expression.

**Figure 6 microorganisms-13-01721-f006:**
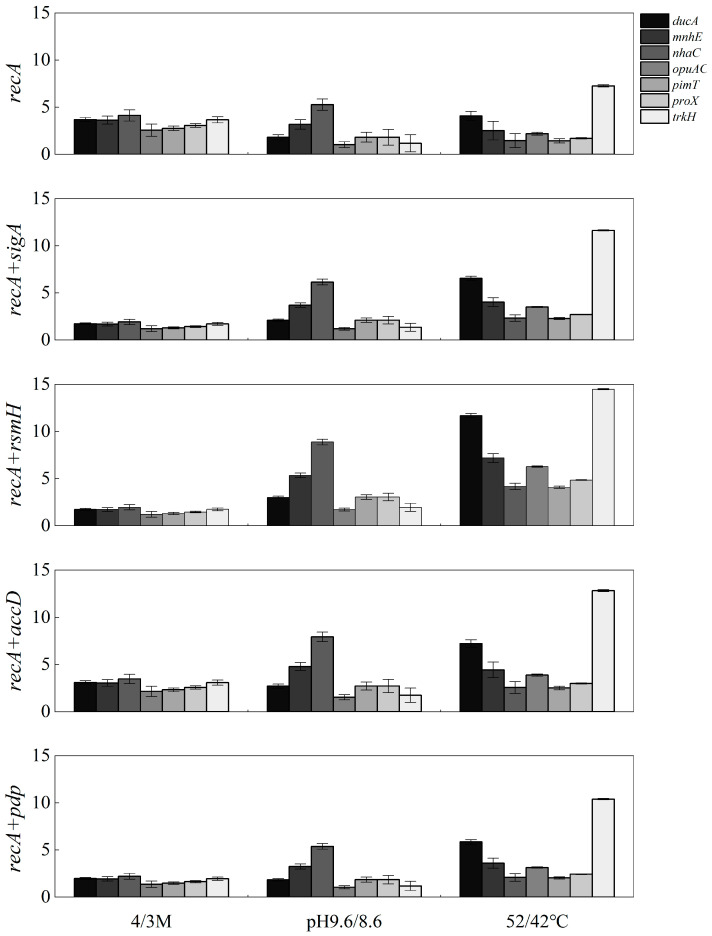
Expression patterns validated via RT-qPCR of the 7 target genes under different conditions normalized against different RGs. Error bars represent means ± SDs.

**Figure 7 microorganisms-13-01721-f007:**
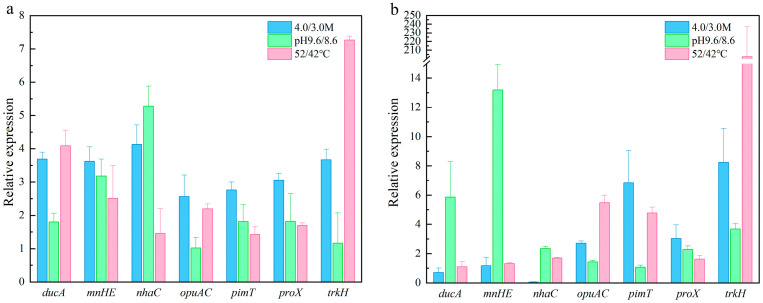
Expression patterns validated via RT-qPCR (**a**) and ddPCR (**b**) of the 7 target genes under different conditions normalized against *recA*.

**Table 1 microorganisms-13-01721-t001:** Slope values of the standard curves and RT-qPCR efficiency of candidate RGs.

Gene Name	Slope	Efficiency (%)	Correlation (R^2^)
*rsmH*	−3.129	108.7	0.996
*pdp*	−3.303	100.8	0.986
*recA*	−3.387	97.3	0.997
*accD*	−3.437	95.4	0.997
*sigA*	−3.513	92.6	0.999
*gyrA*	−3.529	92.0	0.999
*rpoB*	−3.579	90.3	0.999
*dnaK*	−3.451	94.9	0.999

**Table 2 microorganisms-13-01721-t002:** Stability of candidate RGs determined via RT-qPCR according to different analysis parameters.

Method	Rank	Salinity		Temperature		Alkaline pH		All Condition	
		Gene Name	Value	Gene Name	Value	Gene Name	Value	Gene Name	Value
Comparative Ct STDEV	1	* recA *	0.34	* pdp *	2.27	* pdp *	2.03	* recA *	0.41
	2	* sigA *	0.34	* accD *	2.28	* recA *	2.04	* sigA *	0.45
	3	* pdp *	0.36	* sigA *	2.31	* gyrA *	2.08	* dnaK *	0.47
	4	* rsmH *	0.36	* recA *	2.48	* sigA *	2.08	* pdp *	0.48
	5	* rpoB *	0.40	* rsmH *	2.73	* rsmH *	2.15	* accD *	0.48
	6	* gyrA *	0.52	* gyrA *	2.74	* accD *	2.40	* rsmH *	0.53
	7	* dnaK *	0.75	* rpoB *	4.88	* rpoB *	4.61	* gyrA *	0.68
	8	* accD *	0.75	* dnaK *	8.53	* dnaK *	8.36	* rpoB *	0.86
geNorm average expression stability values M	1	* rsmH|recA *	0.11	*pdp|sigA*	0.32	*pdp|recA*	0.12	* recA|sigA *	0.18
	2	*sigA*	0.12	*accD*	0.34	*sigA*	0.21	* pdp *	0.20
	3	*pdp*	0.13	*recA*	0.50	*gyrA*	0.29	*dnaK*	0.28
	4	*rpoB*	0.19	*gyrA*	0.68	* rsmH *	0.36	*accD*	0.33
	5	*gyrA*	0.28	* rsmH *	0.84	*accD*	0.48	* rsmH *	0.37
	6	*dnaK*	0.39	*rpoB*	1.86	*rpoB*	1.51	*gyrA*	0.44
	7	*accD*	0.48	*dnaK*	3.53	*dnaK*	3.22	*rpoB*	0.55
NormFinder stability value SV	1	*recA*	0.05	*accD*	0.17	*recA*	0.50	*recA*	0.15
	2	* pdp *	0.06	* pdp *	0.51	* pdp *	0.51	*dnaK*	0.21
	3	*sigA*	0.06	*sigA*	0.55	*sigA*	0.65	*accD*	0.25
	4	*rsmH*	0.13	*recA*	1.03	*gyrA*	0.70	*sigA*	0.26
	5	*rpoB*	0.14	* rsmH *	1.57	* rsmH *	0.94	* pdp *	0.33
	6	*gyrA*	0.40	*gyrA*	1.95	*accD*	1.45	* rsmH *	0.35
	7	*dnaK*	0.73	*rpoB*	3.76	*rpoB*	3.59	*gyrA*	0.58
	8	*accD*	0.73	*dnaK*	8.36	*dnaK*	8.21	*rpoB*	0.80
BestKeeper SD [±CP]	1	*dnaK*	0.30	*recA*	0.29	*recA*	0.33	*gyrA*	0.62
	2	*gyrA*	0.55	*gyrA*	0.68	*sigA*	0.40	*pdp*	0.71
	3	*rpoB*	0.84	*pdp*	0.74	*pdp*	0.41	*sigA*	0.81
	4	*pdp*	0.97	*sigA*	0.87	*gyrA*	0.61	*recA*	0.85
	5	*recA*	0.99	*accD*	0.97	*accD*	0.66	*dnaK*	0.89
	6	*sigA*	1.01	*rsmH*	1.74	*rsmH*	0.79	*accD*	1.09
	7	*rsmH*	1.06	*rpoB*	2.37	*rpoB*	1.39	*rsmH*	1.19
	8	*accD*	1.54	*dnaK*	5.03	*dnaK*	4.93	*rpoB*	1.28
RefFinder geomean of ranking values	1	*recA*	1.50	*pdp|sigA*	0.32	*recA*	1.19	*recA*	1.41
	2	*pdp*	3.13	*accD*	0.34	*pdp*	1.57	*sigA*	2.21
	3	*sigA*	3.22	*recA*	0.50	*sigA*	2.91	*pdp*	3.31
	4	*rsmH*	3.25	*gyrA*	0.68	*gyrA*	3.72	*dnaK*	3.31
	5	*dnaK*	4.30	*rsmH*	0.84	*rsmH*	5.23	*gyrA*	4.30
	6	*rpoB*	4.40	*rpoB*	1.86	*accD*	5.73	*accD*	4.61
	7	*gyrA*	4.56	*dnaK*	3.53	*rpoB*	7.00	*rsmH*	6.24
	8	*accD*	8.00			*dnaK*	8.00	*rpoB*	8.00

## Data Availability

The original contributions presented in this study are included in the article/Appendix A. Further inquiries can be directed to the corresponding author.

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
