# Peer review of "Quantitative Evaluation of Endogenous Reference Genes for RT-qPCR and ddPCR Gene Expression Under Polyextreme Conditions Using Anaerobic Halophilic Alkalithermophile *Natranaerobius thermophilus"

_microorganisms, 2025, doi:10.3390/microorganisms13081721_

Round 1
Reviewer 1 Report
Comments and Suggestions for Authors
The introduction of the article presents the context and the importance of study. It highlights the stable gene reference need for the quantification of gene expression under poly extreme conditions. The introduction is clear and explains the importance of RGS for the standardization of RT-QPCR data. It also underlines the lack of systematic studies on RGS under poly extreme conditions.
The introduction provides a good overview of previous studies and shortcomings in research, which justifies the need for this study. The objectives are clearly stated: evaluating the stability of RGS under poly extreme conditions and validating their use for the standardization of gene expression.
The methods are described in detail, which allows reproducibility of the study. Growth conditions, RT-QPCR and DDPCR techniques, and statistical analysis are well explained. The selection of eight candidate’s genes is justified by previous studies and genomic information.
The use of several algorithms (Genorm, Normfinder, Bestkeeper, Reffinder) to assess the stability of RGs is a robust approach. The results show good amplification efficiency for candidate RGS, with high R² values, indicating a strong linear correlation. The results of the various analyzes are consistent and show that Rec a is the most stable RG in all the conditions tested. Validation RGs selected by RT-QPCR and DDPCR is well presented and shows good consistency between methods.
The discussion explains well why Rec is the most stable RG and how this is consistent with its biological role in the repair of DNA and homologous recombination. Discussion compares the results with other studies on RGS under extreme conditions, emphasizing the importance of the specific validation of RGS for each organization and condition.
The implications of the study for research on extreme microorganisms and potential applications in microbial biotechnology are well explained. The conclusion sums up the main results well, that Reca is the most stable RG under polyextreme conditions.
The implications for future studies on gene expression in extreme conditions are clearly stated. The conclusion suggests future directions for research, such as the evaluation of the stability of RGs under chronic poly extreme stress and the application to other extreme archaea. A more in-depth discussion on the potential limitations of the study, such as the generalization of results to other organisms or conditions, could be appreciate. A criticism of the potential limitations and challenges for the generalization of the results could also be expected. Additionally, providing more details on the specific stress conditions and their lift to natural surroundings Could enhance The Study's Applicability.
Remarque
Why not have used an RNA-SEQ approach to confirm or deny the results of the study on the stability of reference genes (RGS) under poly extreme conditions? Using RNA-SEQ, you could measure the expression of candidate RGS under the same experimental conditions as those used in the study. This would confirm if Rec a and the other identified RGS are actually stable on the scale of the entire transcriptom. RNA-SEQ provides an overview of gene expression and can reveal variations that could be missed by techniques targeting specific genes, such as RT-QPCR.
Reviewer 2 Report
Comments and Suggestions for Authors
I have reviewed the manuscript entitled “Quantitative evaluation of endogenous reference genes (RGs) for RT-qPCR and ddPCR gene expression under polyextreme using anaerobic halophilic alkalithermophile Natranaerobius thermophilus”. This study evaluates the suitability of eight candidate reference genes to be used in RT-qPCR analysis in anaerobic halophilic alkalithermophile Natranaerobius thermophilus JW/NM-WN-LFT. The manuscript is well presented, and the results are interesting, so in my opinion, it can be accepted after addressing some important issues:
1.- Add the word “conditions” after “polyextreme” in the title.
2.- In introduction (or in Materials and Methods), it would be interesting to include information (identity and function) about the proteins encoded by the genes selected as RGs and those used to measure their expression levels, in order to understand the reasons why they were selected.
3.- Please explain why the study “…..contributes to the broader understanding of gene expression regulation in extremophiles” (final paragraph of the Introduction).
4.- In Materials and Methods, explain the experimental design used to analyze the reliability and accuracy of the selected RGs and the expression of the selected genes, under various temperatures, pHs and salinities simultaneously (combined salt, alkali, and thermal stresses).
5.- Please revise the inconsistencies between Figure 7 and the sentence: Specifically, under salt conditions of 3.0 M and 4.0 M Na⁺, these two genes were upregulated in RT-qPCR but downregulated in ddPCR (Figure 7).
6.- In Discussion, it is not accurate to say that N. thermophilus is “under stress” at the culture conditions used in the study (This finding is consistent with its biological role in maintaining genomic integrity under stress through DNA repair and homologous recombination), since those are the normal conditions for this microorganism.
Reviewer 3 Report
Comments and Suggestions for Authors
In the scientific article entitled " Quantitative evaluation of endogenous reference genes (RGs) for RT-qPCR and ddPCR gene expression under polyextreme using anaerobic halophilic alkalithermophile Natranaerobius thermophilus”, the authors identify reliable endogenous reference genes for use in N. thermophilus. The article provides important and valuable data. However, there are several areas where the manuscript could be improved in terms of clarity and comprehensibility.
Title
- Title – please do not use any abbreviations
Introduction
- The introduction lacks a clear discussion of the research gap. Clearly state what innovation this study brings compared to existing studies.
- The aim of the study should be reformulated to emphasize the novelty of the experiments conducted. The statement “…….contributes to the broader understanding of gene expression regulation in extremophiles” is too far – please rewritten.
Materials and methods
- pH55℃ - what does it mean? In the context that “.. temperatures ranging from 35 to 56℃…”
- “Cultures were incubated at 42°C and 52°C to collect samples under different thermal conditions” – above it was written that “.. temperatures ranging from 35 to 56℃…” – please rewritten to make it clear.
- Please write the value of OD of bacterial inocula.
Results
- Data on RNA integrity number (RIN) and concentration are missing.
- Fig. 3 should be reconstructed – it is difficult to read the data.
Discussion
- Please highlight in particular what the authors have discovered – what is new based on your results.
Conclusions
- Please add a paragraph that will the describe the limitations of the conducted study.
Round 2
Reviewer 3 Report
Comments and Suggestions for Authors
The manuscript can be published in the revised form.